# Effects of the Policy of Physical Education Entrance Examination for Senior High School on Health in Middle School Students

**Jiahui Dong †, Lin Mo †, Yan Shi, Dongsheng Lu \*, Chen Guo, Zicheng Wan and Bingjun Wan \***

Shaanxi Provincial Children and Adolescent Physical Education Research Center, Physical Education College of Shaanxi Normal University, Xi'an 710119, China

\* Correspondence: ludongsheng@snnu.edu.cn (D.L.); bingjunw55@snnu.edu.cn (B.W.);
  Tel.: +86-18792900065 (B.W.)

† These authors contributed equally to this work.

**Abstract: Background:** The policy of the Physical Education (PE) Entrance Examination for Senior High School (PEESHS) is an operable and measurable educational policy proposed by the Chinese government to solve the youth's physical health problems and promote health in middle school students. In recent years, the reform of PEESHS policy has brought youth sports to a new climax, and determining how to achieve the maximum benefit of health promotion with the PEESHS policy is the current focus of the Chinese government, society, schools, and families. The primary purpose was to investigate the health promotion benefits of PE on junior high school students under PEESHS policy and clarify the differences and correlation of overall health, physical fitness, sports participation, social adaptation, and learning facilitation. The secondary aim was to assess the practical value of PEESHS policy implementation on health promotion. **Methods:** The questionnaire of this study was compiled in four steps, and 31 provincial capitals across China were selected as sampling areas, using the convenience sampling method and snowball sampling method, respectively. The number of questionnaires collected was 11,373 (5703 online; 5670 offline), of which 8574 were valid, with an efficiency rate of 75.4%. Ultimately, 8574 students (4199 girls; 4375 boys) were recruited from junior high schools in 31 provinces and municipalities. Data analyses were performed using ANOVA, *t*-test, and Pearson bivariate correlation. Results: The results showed that the PEESHS significantly improved participants' physical fitness, interpersonal relationships, exercise participation, learning efficiency, and psychological health after preparing for PEESHS. Exercise participation and physical fitness showed the most significant positive correlation. Excessive sports intensity was detrimental to health promotion. **Conclusion:** The PEESHS policy has significantly impacted the health of students participating in PEESHS.

**Keywords:** physical education entrance examination for senior high school; junior high school students; health promotion

## 1. Introduction

The growing number of adolescents sedentary, obesity, myopia and chronic diseases have become major problems in various fields around the world, arousing the interest of governments and the public in physical activity (PA). The 2020 China's National Health and Wellness Commission conducted a comprehensive myopia survey on Chinese students, covering 8604 schools in China and screening 2,477,000 students. The results showed that in 2020, the overall myopia rate among Chinese children and adolescents was 52.7% [1]. The 2020 findings show an improved situation compared to 2019, but the high myopia rate among middle and high school students is a significant health challenge. In the "Blue Book for Children: China's Child Development Report (2021)", it is shown that the overweight and obesity rate of primary and secondary school students in China was 15.5% in 2010, which rose to 20.4% in 2014, and continued to rise to 24.2% in 2019. The overweight and

obesity rate of primary and secondary school students increased by 8.7 percentage points from 2010 to 2019, and the overweight and obesity rate of primary, middle, and high school students in 2019 reached 26.2%, 23.1%, and 21.0%, respectively [2].

During adolescence, adequate levels of PA are essential for the development of basic cognitive, motor and social skills and for musculoskeletal, cardiovascular and metabolic health [3]. PA is an important determinant in the prevention and treatment of adolescent obesity and early metabolic risk factors [4,5]. Adolescence obesity is associated with a number of serious health problems, including psychosocial consequences and increases the risk of non-communicable diseases later in life. Adolescence PA levels also tend to be tracked into adulthood, so establishing healthy PA behaviors in adolescence can be beneficial later in life [6]. Meanwhile, a large number of studies have shown that the human and economic costs caused by the lack of PA are also huge. In the UK, Australia and Canada, for example, the direct and indirect costs of treating diseases caused by physical inactivity are £8.2 bn, £1.5 bn and £65.3 bn, respectively. However, these global costs can be avoided by increasing PA. Current public health guidelines for PA call for 60 min or more of moderate-to-vigorous physical activity per day for adolescents [7–9]. The available evidence suggests that most young people in economically developed countries do not meet this criteria and that their PA levels are likely to have declined significantly in recent decades. The expert panel recommended the implementation of policies to increase physical activity among young people through school-based initiatives.

Schools have long been recognized as an effective place to intervene in adolescents' underlying health problems. Between 6 and 12 years of their lives, they spend more than half of their waking hours in school during the school year [10]. This creates a long-term window of opportunity to promote youth participation in physical activity, regardless of their life circumstances. Schools have a role to play in providing health and physical education, which in turn influences attitudes towards lifestyle and contributes to the level of daily physical activity. However, the core mission of schools is learning, so resource constraints and public pressure to optimize academic achievement limit students' opportunities for physical activity on traditional school days. It is in recognition of the particularity of schools in health promotion that the Chinese government, through years of continuous exploration, finally takes "examination" as a compulsory means to promote schools to play a key role, and makes young people realize that if they do not actively participate in sports activities, change their lifestyle, and improve their physical fitness, they are very likely to fail the physical education exam, which will affect their admission to high school.

Recognizing the special nature of schools in enhancing PA, and in order to address the health dilemma of Chinese adolescents due to lack of physical activity, the Chinese government, after years of exploration, finally adopted "examinations" as mandatory tools to strengthen school interventions in youth health issues and developed a series of related policies and systems. The most typical of these policies is the "physical education examinations for students graduating from junior high school and moving up to middle school" (referred to as "PEESHS") [11].

The history of PEESHS policy can be traced back to the late 1980s. After more than 40 years of development, the development form and policy evolution of PEESHS have been explored and deepened in the dynamic development of school sports policy and are now gradually standardized and institutionalized (Table 1). The history of PEESHS policy can be traced back to the late 1980s. After more than 40 years of development, the development form and policy evolution of PEESHS have been explored and deepened in the dynamic development of school sports policy and are now gradually standardized and institutionalized (Table 1).

**Table 1.** The historical development of the PEESHS.

| Time | Content |
|------|---------|
| 1978 | Comrade Deng Xiaoping stressed the importance of a "comprehensive assessment of education" at the National Education Work Conference [12]. |
| 1979 | The Ministry of Education held a meeting in Yangzhou, China, to emphasize the importance of school sports for health and wellness [13]. |
|  | Chongming Middle School in Shanghai, China, makes its first attempt at a physical education midterm. |
| 1980 | Six provinces and cities in China, including Shanghai, Shandong, Henan, Liaoning, Hunan, and Beijing, followed suit with pilot physical education entrance exams. |
| 1981 | The National Conference on School Physical Education and Health summarized the work of the physical education examination points and guided the next step [14]. |
| 1982 | The Department of Physical Education of the Ministry of Education held a "Seminar on Physical Education for Junior High School Graduates", by which time 24 provinces and cities in China had started to conduct physical education examinations [15]. |
| 1983 | The number of provinces and municipalities in China that have implemented physical education exams increased to 34, tending to be implemented nationwide [16]. |
| 1985 | For the first time, the Chinese government explains the "education diversion," and the physical education examination becomes one of the forms [17]. |
| 1987 | China's education philosophy changed to "quality education", and physical education is gradually included in the entrance examination [18]. |
| 1990 | China's State Council promulgated the Regulations on School PE Work, which marked the official inclusion of physical education as a subject on the national entrance examination [15]. |
| 2007 | The State Council of the Central Committee of the Communist Party of China issued Opinions on Strengthening Youth Sports and Enhancing Youth Physical Fitness, affirmed the weight of the physical education examination in the admissions test, and marked the full-scale implementation of the physical education examination [18]. |
| 2012–2020 | In several school sports policies and education policies, the optimization and reform of the PEESHS was considered. |

With the increasing percentage of PEESHS in the total score in recent years, schools, families, students, and society attach more importance to PE. From the perspective of students' exercise participation time in school, junior high school students, especially junior 3 students, have significantly increased participation in physical exercise and sports activities [19]. The rate of 1 h of physical exercise at school in junior 3 students was 42.7%, ref. [20] 12.1% higher than senior 1 school students. From the perspective of the excellent rate of physical fitness standards, junior 3 students were 29.2%, ref. [20] 6.6% higher than senior 1 students. From the research status, with the increasing proportion of physical education scores in examinations, it has attracted the attention of domestic scholars. The author uses CiteSpace5.8R3 (64-bit) software to search the Chinese database with "Physical education examination", "PEESHS", and "physical education additional examination" as the main title (Table 2). The value of PEESHS in improving the participation in PE class and exercise in junior high school students is evident. However, its benefits on health promotion need to be evaluated from a health perspective.

Betweenness represents the attention of the research field, and the higher the value of betweenness, the more attention the research field pays to the relevant content. In Table 2, PEESHS and High School Entrance Examination Physical Education were high-frequency and high betweenness terms, but they were the search terms of this calculation, so they were not given a special description. However, it could be seen from the word list of high frequency, and high betweenness, and related literature that the frequency of "physical education teaching" ranked the third, and the betweenness reached 0.11, indicating that Chinese scholars focus on physical education teaching when studying the field of physical education high school examination. Through the analysis of the research focus, it not only laid a theoretical foundation for the author's follow-up research, but also discovered

the shortcomings of the existing research, such as the lack of research on the setting of examination items, the problems existing in the project, the examination form and the promoting effect on students. Based on the visualization analysis of the database, the author could describe the health promotion of the sports high school examination policy from a comprehensive to a local perspective.

**Table 2.** Information list of high frequency keywords related to sports high school entrance examination.

| Size | Keyword | Frequency | Betweenness |
|------|---------|-----------|-------------|
| 1 | PEESHS | 325 | 0.89 |
| 2 | High school Entrance Examination Physical Education | 183 | 0.61 |
| 3 | Teaching in Physical Education | 105 | 0.11 |
| 4 | The senior school entrance exam | 65 | 0.10 |
| 5 | The influence of PEESHS | 51 | 0.04 |
| 6 | School Physical Education | 50 | 0.10 |
| 7 | Countermeasure | 47 | 0.03 |
| 8 | Current situation | 45 | 0.03 |
| 9 | Sports | 34 | 0.06 |
| 10 | Physical education test | 32 | 0.10 |
| 11 | Reform | 26 | 0.02 |
| 12 | Physical education additional examination | 23 | 0.04 |
| 13 | Project settings | 17 | 0.01 |
| 14 | Junior high school Physical Education | 16 | 0.04 |
| 15 | Physical Training | 13 | 0.01 |
| 16 | Influence factors | 12 | 0.01 |
| 17 | Strategy | 11 | 0.01 |
| 18 | Physical quality | 10 | 0.03 |

As a national education policy to promote junior high school students to actively participate in sports, PEESHS aims to improve the physical health level of junior high school students and relieve the pressure of study and life. Therefore, in order to explore the health promotion effect of the sports high school examination policy, it was necessary to measure the health promotion effect of the policy from the physical, psychological, social adaptation, moral and other perspectives under the learning and living situation of junior middle school students. Finally, the correct application of PEESHS policy could improve various health problems of adolescents, avoid health risks, and promote their all-round healthy growth.

## 2. Materials and Methods

### 2.1. Participants

This study focused on the direct beneficiary participants of PEESHS, referred to as direct stakeholders. From the perspective of the main groups implementing and participating in the PEESHS, the stakeholders coexist at three levels: the first level is the state-centered national interest, which is the top-level design, and the department that sets the general policy of PEESHS, including the Ministry of Education, and the General Administration of Sports of China; the second level is the local interest, which is the department that implements PEESHS policy and sets the local specific program, i.e., the local education administration; the third level is the individual interests, which are those who specifically implement the policies and programs of PEESHS, including students, PE teachers, parents, school administrators, and various PEESHS training institutions (Figure 1). The first layer is the most significant beneficiary of implementing the PEESHS policy. The realization of its goals needs to be supported by the second and third layers. Only when the interests of the second and third layers are guaranteed to the greatest extent in the policy implementation can the standard implementation of the policy be ensured [21] and the value of the PEESHS policy be fully reflected.

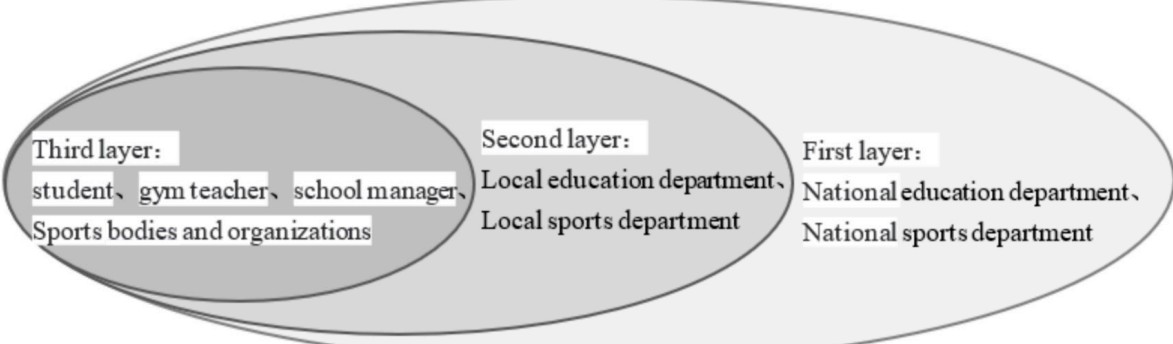

**Figure 1.** PEESHS stakeholder relationship diagram.

This study investigated the health promotion of PEESHS from the perspective of students (third layer). In order to obtain the maximum diversity of our primary study population and to include more students engaged in PEESHS in this study, the inclusion criteria for participants were as follows: (1) middle school students who are attending PEESHS, students in junior 3; (2) middle school students preparing for PEESHS, i.e., students in junior 1 and 2; (3) senior high school students who have just participated in PEESHS.

*2.2. Questionnaires*

The main research method of this study was investigation, including an interview and questionnaire. In the early studies, a large number of documents were read to find a scale that was consistent with the direction of the study, but we failed to find it. Therefore, to ensure the proper conduct of the study, the research team began to develop the scale based on the existing studies. This study followed the basic steps of questionnaire design and the principles of survey scale design. In the first step, the purpose, investigation direction, and scope of this study were clarified, and the structure of the scale was planned rationally. In the second step, the research team constructed the logical framework and the overall framework of the survey questions by compiling the interview results and the literature. In the third step, the research team formally wrote the questions and iteratively revised the expression form to adjust the scale's overall layout. In the fourth step, experts in related fields were invited to review and revise the questionnaires. In the fifth step, the research team conducted a pre-investigation, analyzed the questionnaires returned from the pre-investigation, and revised the questionnaires according to the results. In the sixth step, experts were invited again to review and finalize the questionnaire scale for this study.

2.2.1. Questionnaire Design Process

(1) Literature collection and classification. This study included the keywords of PEESHS, physical education in the secondary school examination, physical education subject in the academic level test, physical education examination score, physical education examination content, and physical education examination. The literature was retrieved from China Knowledge Network, China Science and Technology Journal Database, WAN FANG DATA, Web of Science, and other databases. (2) Interviews and group discussions. Experts in physical education and school physical education, teachers, and junior high school students were invited for interviews. (3) Questionnaire design and dimensional conception. This study was based on literature review and classification, combined with the interview and dimensional division of the domestic PEESHS-related research, to construct questionnaire dimensions consistent with this study. (4) After the questionnaire's initial preparation, the returned questionnaire's reliability and validity were analyzed after the first round of expert review and pre-research. Some questions were revised after the analysis, and the second pre-research was conducted. The questionnaire was analyzed and revised again, concentrated on the realistic problems of this study, and the questionnaire was finalized.

### 2.2.2. Questionnaire Determination

The scale developed for this study focused on individual changes in junior high school students after the PEESHS intervention (or a period of preparation). It measured PEESHS's health promotion effect on junior high school students.

The questionnaire for this study consisted of two parts: the first part, basic personal information; and the second part, the current status of PEESHS on health promotion. The basic information contains questions on demographic characteristic variables.

The section on the current status of PEESHS on health promotion contained 26 questions divided into five dimensions (See Table S1 for details). Participants were asked to rate their degree of conformity using a 5-point Likert scale for each statement, from 1 (strongly disagree) to 5 (strongly agree). The current status of PEESHS on the health promotion questionnaire ranged from 26 to 130, with higher scores reflecting a greater degree of conformity. The internal consistency for the present study was 0.84, and physical fitness ($\alpha$ = 0.86), psychological health ($\alpha$ = 0.82), interpersonal relationships ($\alpha$ = 0.84), exercise participation ($\alpha$ = 0.82), and learning efficiency ($\alpha$ = 0.82) have shown good levels of reliability. Since the scale was completed under the guidance of experts, the scale was evaluated using expert assessment and pre-research methods for validity testing.

### 2.3. Questionnaire Distribution

Since it is difficult to investigate the entire sample, this study finalized the scope of the research based on the actual situation of the research participants and the research group, supported by relevant literature. The research team selected 31 provincial capital-level cities across the country as the sampling area. It used convenience and snowball sampling methods online and offline for specific operations. This method was convenient, fast, and could accumulate many samples after a short period. However, there was an uncontrollable nature of sampling. The homeopathic bias of the sample was controlled within a specific range by process optimization and statistical tests [22]. The number of questionnaires collected was 11,373 (5703 online; 5670 offline), of which 8574 were valid, with an efficiency rate of 75.4%. Ultimately, 8574 students (4199 girls; 4375 boys) were recruited from junior high schools in 31 provinces and municipalities. The study protocol was reviewed and approved by the Project Decision Advisory Committee of the General Administration of Sports of China in 2021 (No. 2021-B-18). Before participating in this study, participants were informed that they were recruited voluntarily, that their data was collected anonymously, and that they could withdraw from the study at any time. The participants or their parents provided signed informed consent. For the offline survey, students completed the questionnaire at school (duration: 15–22 min), and the online questionnaires were completed during students' time outside of class (duration 13–18 min).

### 2.4. Statistical Analysis

The results were analyzed using the Statistical Package for the Social Sciences (SPSS version 26.0). The normality and reliability coefficient ($\alpha$) of variables were screened to ensure the authenticity and effectiveness of the recovered data. Descriptive statistics were used to analyze the basic information of participants, including the total number of participants, the number of different genders, the number of different grades, and the number of different schools. A *t*-test was used to compare the differences between men and women. Analysis of variance (ANOVA) was used to compare the differences between students in different grades and schools in different dimensions. The correlation between each dimension was observed by calculating the Pearson bivariate correlation coefficient.

## 3. Results

### 3.1. Demographic Characteristics of Participants

Table 3 shows the descriptive statistics of the participants (*n* = 8574). The mean age of the participants was 13.46 years ($\pm$SD 1.38). These participants included students in junior 1 (1983 students, 23.1%), junior 2 (1771 students, 20.7%), junior 3 (2440 students, 28.5%),

and senior 1 (2380 students, 27.8%). Participants came from different levels of schools, such as provincial capital middle schools (1851 students, 21.6%), urban middle schools (3140 students, 36.6%), county middle schools (1434 students, 16.7%), and village and town middle schools (2149 students, 25.1%).

**Table 3.** Characteristics of the study samples (*n* = 8574).

| Socio-Demographics | *n* | % or Mean |
|---|---|---|
| Boys | 4375 | 51.00 |
| Girls | 4199 | 49.00 |
| Junior 1 | 1983 | 23.10 |
| Junior 2 | 1771 | 20.70 |
| Junior 3 | 2440 | 28.50 |
| Senior 1 | 2380 | 27.80 |
| Provincial capital middle school | 1851 | 21.60 |
| Urban middle school | 3140 | 36.60 |
| County middle school | 1434 | 16.70 |
| Village and town middle school | 2149 | 25.10 |

### 3.2. Common Method Deviation Test

Since the data were collected through questionnaires, in order to eliminate common method bias, we used a tool to control for common method bias [23–25] to test common method bias [26]. The results showed 14 factors with eigenvalues greater than one, and the variance explained by the first factor was 34.71%, which is less than 40% of the critical standard criterion, proving that the bias of the common methods is not significant [25–27].

### 3.3. Differences in Different Groups of Sex, Grade, and School

Table 4 uses an independent samples t-test to determine the differences between boys and girls according to the five dimensions. The results show that boys scored significantly higher than girls in the physical fitness dimension ($p < 0.01$). The scores of girls in the psychological health and learning efficiency dimension are significantly higher than those of boys ($p < 0.01$).

**Table 4.** Sex difference of health promotion in PEESHS.

| Dimensions | Boys | | Girls | | t | p |
|---|---|---|---|---|---|---|
| | M | SD | M | SD | | |
| Physical fitness | 3.76 | 0.77 | 3.63 | 0.76 | 7.92 | < 0.01 |
| Psychological health | 4.03 | 0.81 | 4.10 | 0.80 | −3.88 | < 0.01 |
| Interpersonal relationship | 3.93 | 0.97 | 3.95 | 0.94 | −0.91 | 0.36 |
| Exercise participation | 3.89 | 1.01 | 3.92 | 1.01 | −1.39 | 0.16 |
| Learning efficiency | 4.05 | 1.00 | 4.17 | 0.95 | −5.74 | < 0.01 |

Table 5 uses one-way ANOVA to determine the differences in five dimensions among the four grades (junior 1, junior 2, junior 3, and senior 1). The results showed that junior 3 students had the highest scores in physical fitness and psychological health. In contrast, senior students had the highest scores in interpersonal relationships, exercise participation, and learning efficiency. In addition, the four grades showed significant differences in five dimensions, as shown in Figure 1.

**Table 5.** Grade difference of health promotion in PEESHS.

| Dimensions | Grade | | | | | | | | F | *p* |
| | Junior 1 | | Junior 2 | | Junior 3 | | Senior 1 | | | |
| | M | SD | M | SD | M | SD | M | SD | | |
|---|---|---|---|---|---|---|---|---|---|---|
| Physical fitness | 3.74 | 0.79 | 3.55 | 0.78 | 3.76 | 0.68 | 3.67 | 0.79 | 30.48 | <0.01 |
| Psychological health | 3.89 | 0.78 | 3.95 | 0.80 | 4.20 | 0.72 | 4.14 | 0.86 | 71.62 | <0.01 |
| Interpersonal relationship | 3.83 | 0.97 | 3.94 | 0.96 | 3.98 | 0.93 | 3.99 | 0.96 | 12.63 | <0.01 |
| Exercise participation | 3.82 | 0.98 | 3.70 | 1.02 | 3.94 | 0.97 | 4.07 | 1.02 | 50.79 | <0.01 |
| Learning efficiency | 3.89 | 1.07 | 4.10 | 0.98 | 4.15 | 0.87 | 4.27 | 0.97 | 55.31 | <0.01 |

The difference between grades showed in Figure 2. LSD was used to make a pairwise comparison between grade groups and visualized for multiple comparisons using the alphabetical marker method (a, b, c, d). The difference was not statistically significant if the two columns had the same letter. The difference was statistically significant if there were no common letters between the two columns.

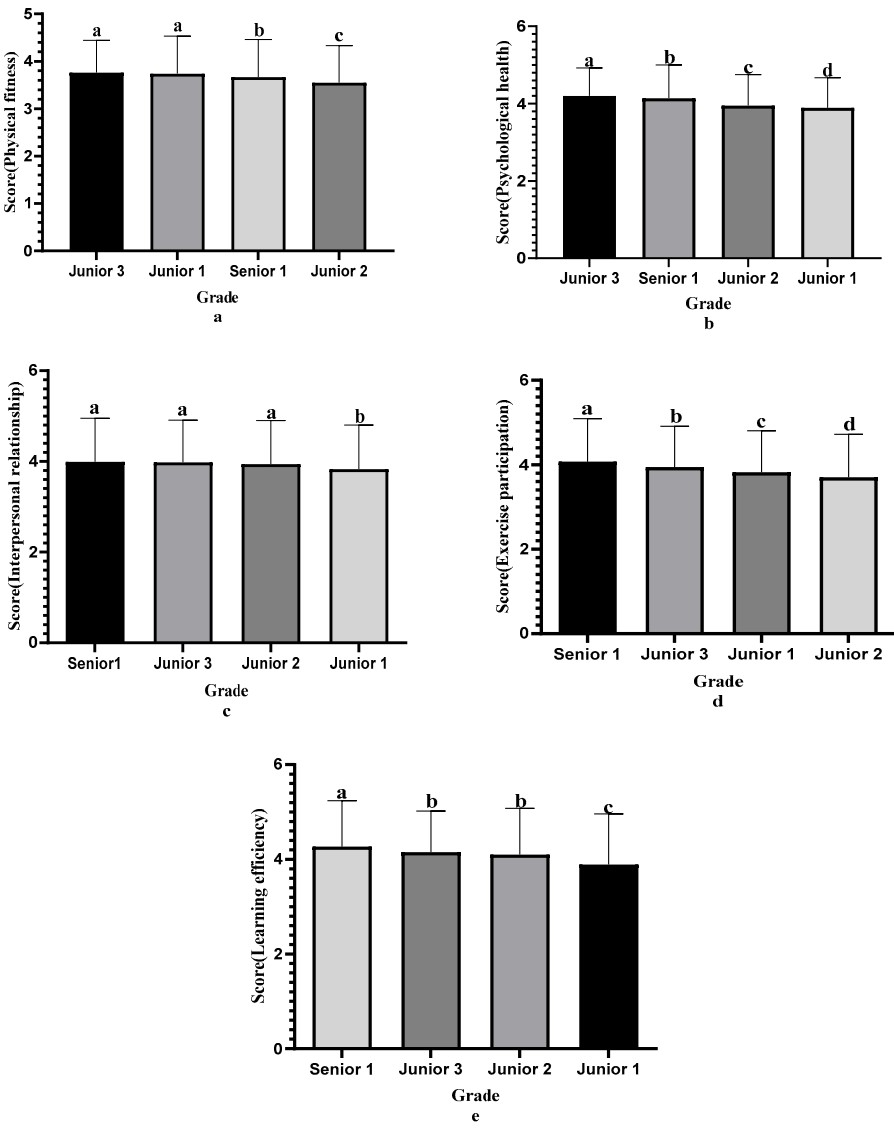

**Figure 2.** Analysis of variability between grades after PEESHS intervention. (**a**): Graph of grade differences in Physical fitness dimensions; (**b**): Graph of grade differences in Psychological health dimensions; (**c**): Graph of grade differences in Interpersonal relationship dimensions; (**d**): Graph of

grade differences in Exercise participation dimensions; (**e**): Graph of grade differences in Learning efficiency dimensions. NOTE: a, b, c, d stands for Alphabetic Marking Method. The difference is not statistically significant if there is the same letter between two columns. The difference is statistically significant if there are no common letters between the two columns.

The results showed the following: in the dimension of physical quality, there was no difference between junior three and junior one students, but there were highly significant differences between junior two and senior one students. In the dimensions of psychological health and exercise participation, all four grades showed highly significant differences. In the dimension of interpersonal relationships, there was no difference between junior two, junior three, and senior one students, but there was a very significant difference between junior one students. In the dimension of learning efficiency, there was no difference between junior two and junior three students. However, there was a very significant difference between junior and senior students.

Table 6 used one-way ANOVA to determine the differences in five dimensions among provincial capital middle school, urban middle school, county middle school, village, and town middle school. The results showed that urban middle school students had the highest physical fitness and exercise participation scores. In contrast, county middle school students scored the highest in interpersonal relationships, psychological health, and learning efficiency. In addition, the four types of school grades showed significant differences in five dimensions, as shown in Figure 3.

**Table 6.** Schools difference of health promotion in PEESHS.

| Dimensions | School | | | | | | | | F | *p* |
|---|---|---|---|---|---|---|---|---|---|---|
| | Provincial Capital Middle School | | Urban Middle School | | County Middle School | | Village and Town Middle School | | | |
| | M | SD | M | SD | M | SD | M | SD | | |
| Physical fitness | 3.67 | 0.73 | 3.72 | 0.76 | 3.65 | 0.76 | 3.69 | 0.80 | 3.45 | <0.01 |
| Psychological health | 4.04 | 0.77 | 4.11 | 0.81 | 4.15 | 0.76 | 3.97 | 0.85 | 18.15 | <0.01 |
| Interpersonal relationship | 3.94 | 0.92 | 3.95 | 0.98 | 4.08 | 0.91 | 3.83 | 0.98 | 20.74 | <0.01 |
| Exercise participation | 3.90 | 0.86 | 4.00 | 1.02 | 3.94 | 1.00 | 3.75 | 1.10 | 27.08 | <0.01 |
| Learning efficiency | 4.19 | 0.86 | 4.18 | 0.98 | 4.25 | 0.84 | 3.87 | 1.11 | 62.34 | <0.01 |

For the difference between schools shown in Figure 3, LSD was used to make a pairwise comparison between school groups and visualized for multiple comparisons using the alphabetical marker method (a, b, c, d).

The results showed the following: in the dimension of physical quality, there was a significant difference between urban middle school and county middle school students, but there was no significant difference between the other two kinds of schools. In the dimensions of psychological health, interpersonal relationship, and learning efficiency, there was no difference between students from provincial middle schools and urban middle schools. However, there was a significant difference between students from county middle schools and village and town middle schools. In the dimension of exercise participation, there was no difference between provincial and county middle schools. However, there was a significant difference between urban middle schools and village and town middle schools.

Students of different sexes, grades, and schools generally scored higher than 3.5 on the degree of health promotion in PEESHS, showing a high degree of recognition.

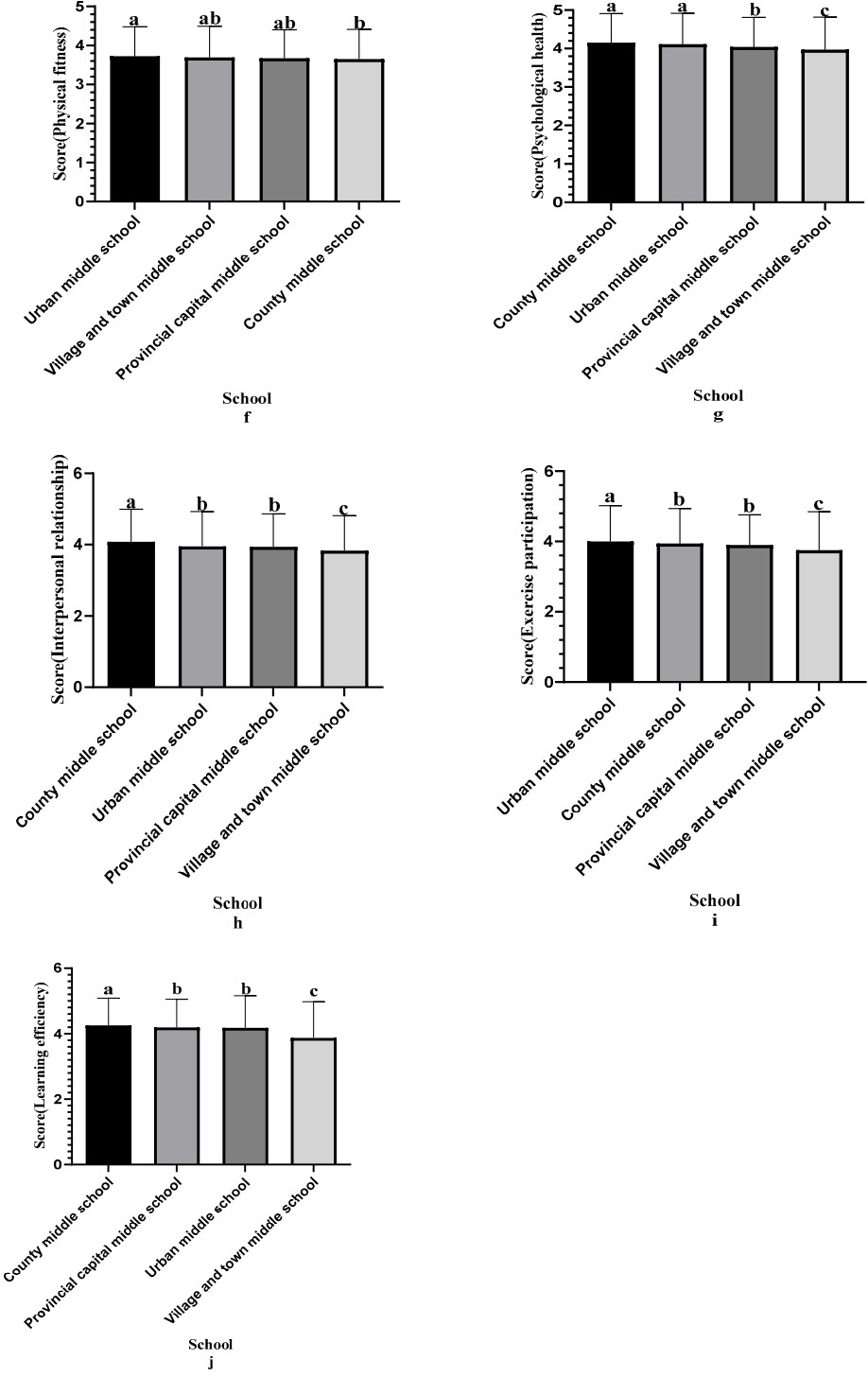

**Figure 3.** Analysis of variability between schools after PEESHS intervention. (**f**): Graph of school differences in Physical fitness dimensions; (**g**): Graph of school differences in Psychological health dimensions; (**h**): Graph of school differences in Interpersonal relationship dimensions; (**i**): Graph of school differences in Exercise participation dimensions; (**j**): Graph of school differences in Learning efficiency dimensions. NOTE: a, b, c, d stands for Alphabetic Marking Method. The difference is not statistically significant if there is the same letter between two columns. The difference is statistically significant if there are no common letters between the two columns.

*3.4. Association between PEESHS-Related Variables*

The correlation coefficient (r) for each variable of the PEESHS is shown in Table 7. Statistically, significant correlations are reported between the PEESHS on physical fitness, psychological health, exercise participation, interpersonal relationships and learning efficiency. Based on the results of the variability and correlation of PEESHS on health status and physical changes, it can be concluded that PEESHS was positively associated with health promotion.

**Table 7.** Intercorrelations among variables for the total sample (*n* = 8574).

| Dimensions | M | SD | Physical Fitness | Psychological Health | Interpersonal Relationship | Exercise Participation | Learning Efficiency |
|---|---|---|---|---|---|---|---|
| Physical fitness | 3.69 | 0.76 | 1 | | | | |
| Psychological health | 4.06 | 0.81 | 0.66 ** | 1 | | | |
| Interpersonal relationship | 3.94 | 0.96 | 0.16 | 0.37 ** | 1 | | |
| Exercise participation | 3.91 | 1.01 | 0.36 ** | 0.52 ** | 0.47 ** | 1 | |
| Learning efficiency | 4.12 | 0.98 | 0.31 ** | 0.79 ** | 0.24 ** | 0.69 ** | 1 |

NOTE: M, Mean; SD, Standard Deviation; ** $p < 0.01$.

**4. Discussion**

Through the investigation of the health status of PEESHS participants, it was found that PEESHS significantly improved the physical fitness, interpersonal relationships, exercise participation, learning efficiency, and psychological health of participants. Among them, exercise participation and physical fitness showed the most significant positive correlation. However, excessive exercise intensity was not conducive to promoting health as a whole, and the effectiveness of the PEESHS as an educational tool to improve teenagers' health and promote active participation in sports was significant. Meanwhile, it also proved the effectiveness of the Chinese government's implementation of PEESHS in addressing the challenges of teenagers' health issues.

Physical exercise is the most substantial factor for long-term health promotion, followed by medical treatment. The former ensures the body's healthy functioning and prevents diseases from harming health, while the latter treats diseases through various drugs and means to accomplish health goals. Physical activity is meant to prevent diseases before they happen.

Numerous studies have shown that health is closely related to lifestyle and that a poor lifestyle with a lack of physical activity is the leading cause of national illness and death [28]. The implementation of the PEESHS policy is precisely intended to promote the development of physical education in schools, with the essential aim of promoting students' active participation in exercise and enhancing their health condition. The results showed that the participants gradually developed the habit of active participation in exercise in preparation for PEESHS, which was reflected in a significant increase in exercise scores of students and parents. The most significant differences in physical fitness and exercise participation were found among the participants in the provincial capital and urban secondary school, implying that the participants in urban areas were previously physically inactive and in poor health. After PEESHS, physical activity and physical fitness were significantly increased, and academic stress was released. These changes enhanced physical fitness and improved the participants' health, as evidenced by the analysis of differences between participants in different grades, with junior three and senior one students who experienced the complete PEESHS process perceiving the PEESHS health promotion as highly significant. Chuan-Sheng Dong mentioned that physical activity could help people build a more active lifestyle and improve their overall health. Our study results showed similar characteristics, i.e., students significantly enhanced physical fitness, exercise participation, interpersonal relationships, and overall health. Implementing the PEESHS policy

may affect participants' health promotion, which confirms that physical education is a crucial way to improve health [20].

At the same time, too much exercise could be harmful to health. Many studies or institutions [29–33] recommend that people should exercise at moderate and higher (high) intensities for a certain amount of time per week (or per day), with a predominance of moderate intensity (e.g., at least 5 d per week is recommended). The American College of Sports Medicine, the Centers for Disease Control and Prevention, the American Heart Association, and relevant organizations in China have successively suggested a range of moderate and higher exercise intensities. For example, the moderate and appropriate high-intensity exercise heart rates identified by the American Heart Association [34] are 50% to 70% and 71% to 85% of the individual's maximum heart rate, respectively. The moderate exercise intensity proposed by the American College of Sports Medicine [31] and the "Healthy China Action (2019–2030)" [35] promulgated in China is 64% to 76% of the maximum heart rate. The American College of Sports Medicine also uses the reserve heart rate (HRR) for intensity grading, where moderate intensity is 40% HRR [33]. In practice, moderate intensity is also usually 60% to 70% of the maximum heart rate. Moderate and higher intensities of physical activity can improve cardiopulmonary function and improve the health of the respiratory and cardiovascular systems [36–38]. The health effects associated with moderate-to-high intensity are also related to metabolism, the musculoskeletal system, cancer, and functional health [29]. However, some of these effects may also result from exercises such as solid muscle and bone activity (not directly related to heart rate intensity) [39].

The definition of health in social semantics is usually cited from the World Health Organization's definition of health, which has the most extensive and far-reaching influence on health-related fields in China. As early as 1948, the World Health Organization proposed a definition of health that is still in use today: "Health is a state of complete physical, mental and social well-being and not merely the absence of disease or infirmity." Improving health through exercise is one of the most effective avenues available today. For example, recreational exercise makes people relaxed and happy. Psychological stress is relieved, and the body is exercised. In the case of students, the rapid development of society, the accelerated pace of life, and the increasing pressure of study have led to health problems, not only physical health problems but also mental health problems [40–46]. The health promotion function of exercise can respond to the needs of human mental health problems, making the health promotion function of sports socialized and expanded. This social development means that more and more people rely on the function of exercise in health promotion to improve their health [47–50]. In the face of the current adolescent health dilemma, the health promotion benefits of exercise should be examined in multiple dimensions. Through the policy tool of PEESHS, the theoretical value of exercise should be raised. The value implication of exercise and physical education should be manifested, thus achieving adolescent health governance and alleviating the Chinese government's and society's adolescent health concerns.

Our study provides results on the health promotion of PEESHS through participant scores in different PEESHS preparation conditions. However, we acknowledge several limitations of our studies. First, the students selected in this study defaulted to having begun preparing for PEESHS, ignoring the time students were preparing for it. For example, for junior one and junior two students, PEESHS is administered throughout the three years of junior high school in some regions. In others, the exam is administered uniformly in the third year. Whether differences in the format and timing of the exam affect the health promotion of PEESHS is still unclear. Second, participants came from all over China, including adolescent athletes, whose data may bias the study results to a small degree. Therefore, the results of this study have some limitations in responding to the health promotion of PEESHS.

Due to the above limitations, more improvements could be made in future studies. First, different preparation time groups (e.g., 0 months, 1–6 months, 7–12 months,

13–18 months, 19–24 months or more) could be added for more detailed data comparisons to draw robust conclusions and reduce the effect of non-controlling variables on the data results. Second, future studies could consider recruiting more middle or high school students (excluding youth athletes) from provinces and municipalities with different economic characteristics to reduce the individualization of study results.

## 5. Conclusions

This study revealed the health-promoting effects of PEESHS, mainly in terms of participants' scores on physical fitness, psychological health, exercise participation, interpersonal relationships, and learning efficiency dimensions of scores. Additionally, sex, grade level, and school are essential related factors. As experts in physical education and health have argued, the health function performed by physical education is a comprehensive promotion from physical, psychological, and social perspectives. Overall, physical fitness and exercise participation have the strongest correlation to health promotion for students. The effect of PEESHS was more robust in girls than boys and was more pronounced among students in provincial and urban secondary schools. Junior three and senior one students were most optimistic about the health promotion value of PEESHS. In addition, different sexes, grades, and schools had different performance levels in PEESHS implementation, with similarly different levels of variation in health promotion outcomes. Finally, exercise participation is a significant factor influencing health promotion, the intensity of which cannot be practically quantified in student sports. Low intensities have insignificant effects on health promotion, and excessive intensities could threaten health and thus reduce the effectiveness of health promotion.

**Supplementary Materials:** The following supporting information can be downloaded at: https://www.mdpi.com/article/10.3390/su15021701/s1, Table S1: List of scores for each dimension item.

**Author Contributions:** B.W. and J.D. conceived of the study, and participated in its design and adjustment, and conceptualized the study; J.D. and L.M. participated in methodological analysis; Y.S. and Z.W. performed statistical analysis; J.D. and C.G. organized the survey; J.D. and L.M. organized the data and participated in the writing—original draft preparation; J.D., D.L. and B.W. organized writing—review and editing; B.W and D.L. supervised and administered the study. All authors have read and agreed to the published version of the manuscript.

**Funding:** This study was funded by the Decision-making Consultation Project of the General Administration of China, Project No.2021-B-18.

**Institutional Review Board Statement:** This study was conducted according to the guidelines of the Declaration of Helsinki and approved by the Academic Committees of Shaanxi Normal University in 2020 (NO: 202116008).

**Informed Consent Statement:** Informed consent was obtained from all subjects involved in the study.

**Data Availability Statement:** Please contact the corresponding authors to obtain the data.

**Acknowledgments:** We thank the Shaanxi Provincial Children and Adolescent Physical Education Research Center members. We also acknowledge the schools and students for participating in the research investigation.

**Conflicts of Interest:** The authors declare no conflict of interest.

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
