# Peer review of "Effects of the Policy of Physical Education Entrance Examination for Senior High School on Health in Middle School Students"

_sustainability, doi:10.3390/su15021701_

Round 1

Reviewer 1 Report

I have read the article and have a few suggestions to the authors: 

1. References are not cited correctly in the text. 

2. The introduction does not give enough information about what the authors want to do in the article, what gap from the literature the study covers and especially the coverage of potential results in the literature. 

3. From the material and methods section the actual presentation of the questionnaire or at least its content is missing. Moreover, the description part of the statistical analysis (section 2.4) is treated very briefly. Here I suggest the authors to elaborate a bit on how to apply the statistical methods to the data obtained from the questionnaire. 

4. The research methodology is not very clearly defined. 

5. I don't really understand what the authors of this study want to prove, what are they bringing new? It has been known for a long time that exercise, whether it is done in an organised way or not, contributes substantially to improving the quality of life, with all that this entails. 

6. I suggest the authors to study and cite more works from the literature and from other areas not only from China. They should also compare their results with those of similar studies conducted in other parts of the world. 

Author Response

1.References are not cited correctly in the text.

Thank you for pointing this out. The format of the references has been readjusted.

2.The introduction does not give enough information about what the authors want to do in the article, what gap from the literature the study covers and especially the coverage of potential results in the literature.

Agreed. The introduction part has been revised to expand the scope of references, and clear research background, and research significance.

3.From the material and methods section the actual presentation of the questionnaire or at least its content is missing. Moreover, the description part of the statistical analysis (section 2.4) is treated very briefly. Here I suggest the authors to elaborate a bit on how to apply the statistical methods to the data obtained from the questionnaire.

Thank you for this insightful comment. The application of statistical methods is described in detail in Part 2.4.

4.The research methodology is not very clearly defined.

Agreed. As for the research method, it was not explicitly proposed in the previous version of the text. After modification, it has been clearly proposed in the article to adopt the survey method, including the interview method and the questionnaire, in which the interview is the premise of the scale preparation.

5.I don't really understand what the authors of this study want to prove, what are they bringing new? It has been known for a long time that exercise, whether it is done in an organised way or not, contributes substantially to improving the quality of life, with all that this entails.

Agreed. On this issue, the author agrees with the expert extremely much. However, from the perspective of China's national conditions, due to high scholastic stress, students spend all their time completing homework after class and have little time for PA, which leads to physical health problems for students due to the lack of PA. The Chinese government realized the seriousness of this problem, and to solve this problem, taking a compulsory means, it is PEESHS. The purpose of this study is to research whether the implementation of the PE high school examination policy has a positive effect on students' health.

6.I suggest the authors to study and cite more works from the literature and from other areas not only from China. They should also compare their results with those of similar studies conducted in other parts of the world.

This suggestion is excellent. It is true that the authors cover very few foreign articles on physical education exams in article, and this will be strengthened in subsequent studies.

Reviewer 2 Report

Thanks for submitting the paper for review. It was a pleasure to read your manuscript. However, I do have some questions and comments toward the work.

1. Based on section 2.2.1, the questionnaire was designed with based on literature and interviews. My question is was the questionnaire updated or adjusted based on previous existing surveys or completely new items created by the researchers? If it was adjusted or based on other existing questionnaire, please specify.

2. It was great to provide the reliability of each factor. But could you please include more detailed information about each factor, such as sample items and the number of items for each factor.

3. It was concluded in the paper that PEESHS was significantly improved students' health status, however, from your research design, I only see gender difference, grade level difference and school location difference among each factors of the questionnaire. What's the connection between such difference and significant impact of PEESHS on health status? 

4. Since the difference of gender, location and grade level was examined for each factor, also each factors are significantly correlated with overall high internal consistency, why not analyze the overall difference?

5. It's a newly developed instrument, although internal consistency was conducted, I suggest that with this large of sample size, you can try to run EFA, CFA and measurement invariance to confirm the factor structure of the instrument for future application. 

Author Response

1.Based on section 2.2.1, the questionnaire was designed with based on literature and interviews. My question is was the questionnaire updated or adjusted based on previous existing surveys or completely new items created by the researchers? If it was adjusted or based on other existing questionnaire, please specify.

Thank you for pointing this out.This study tried to find a scale consistent with the direction of this study by reading a large amount of literature, but failed to find one. Therefore, to ensure the proper conduct of the study, the research team began to develop the scale based on the existing studies.

2.It was great to provide the reliability of each factor. But could you please include more detailed information about each factor, such as sample items and the number of items for each factor.

Thanks. Regarding this suggestion, the authors have included it as supplementary material in the article.

3.It was concluded in the paper that PEESHS was significantly improved students' health status, however, from your research design, I only see gender difference, grade level difference and school location difference among each factors of the questionnaire. What's the connection between such difference and significant impact of PEESHS on health status?

Thanks. The results showed that the PEESHS has a significant effect on the health of students. However, from the perspective of gender, grade and school location, there were differences among all factors. In terms of gender, it indicated that the improvement effect of PEESHS on boys and girls was different. The improvement effect of boys on physical fitness, psychological health and learning efficiency was significantly higher than that of girls, while the improvement effect on interpersonal relationship and exercise participation was the same. In terms of grades, the junior 3 and senior 1 who have completed the high school physical education examination score higher than the junior 1 and junior 2 students who are preparing for the high school physical education examination, indicating that the improvement effect of the high school physical education examination on the junior 3 and senior 1 students is more significant. The reason was that the junior 3 and senior 1 students spend more time on the high school physical education examination, so the effect was more obvious. In terms of school location, Urban middle school and County middle school generally scored higher than the other two types of schools, indicating that students in Provincial capital middle school have good resources and more ways to achieve high scores, and working hard on sports training is just one of the ways; while students in Village and town middle school are relatively backward in terms of resources due to the local economic situation and family economic conditions, which include insufficient resources for sports, so the ways to exercise are relatively single.

4.Since the difference of gender, location and grade level was examined for each factor, also each factors are significantly correlated with overall high internal consistency, why not analyze the overall difference?

Thanks. It is well worth thinking about this issue raised by the experts. After recovering the data, the authors found that the overall high scores of the students indicate a high level of acceptance of the PE midterm policy. It is based on the overall scores that a detailed description was not made but rather chose to analyze the internal variability in terms of different genders, grades, and schools to obtain the improvement of the PEESHS for different genders, grades, and schools.

5.It's a newly developed instrument, although internal consistency was conducted, I suggest that with this large of sample size, you can try to run EFA, CFA and measurement invariance to confirm the factor structure of the instrument for future application.

Thanks. Since the scale was completed under the guidance of experts, the scale was evaluated using expert assessment and pre-research methods for validity testing.The suggestions you put forward will be strengthened in the future research.

Round 2

Reviewer 1 Report

From my point of view the paper can be accepted in the present form.